# Multidetector CT Imaging Biomarkers as Predictors of Prognosis in Shock: Updates and Future Directions

**DOI:** 10.3390/diagnostics13132304

**Published:** 2023-07-07

**Authors:** Tullio Valente, Giorgio Bocchini, Candida Massimo, Gaetano Rea, Roberta Lieto, Salvatore Guarino, Emanuele Muto, Ahmad Abu-Omar, Mariano Scaglione, Giacomo Sica

**Affiliations:** 1Department of Radiology, Monaldi Hospital, Azienda dei Colli, 80131 Naples, Italygsica@sirm.org (G.S.); 2Department of Radiology, Vancouver General Hospital, 899 W 12th Avenue, Vancouver, BC V5Z 1M9, Canada; 3Department of Radiology, James Cook University Hospital, Middlesbrough TS4 3BW, UK; 4Department of Medicine, Surgery and Pharmacy, University of Sassari, 07100 Sassari, Italy

**Keywords:** cardiogenic shock, computed tomography, contrast layering, venous pooling, hypovolemia, hypovolemic shock complex

## Abstract

A severe mismatch between the supply and demand of oxygen is the common sequela of all types of shock, which present a mortality of up to 80%. Various organs play a protective role in shock and contribute to whole-body homeostasis. The ever-increasing number of multidetector CT examinations in severely ill and sometimes unstable patients leads to more frequently encountered findings leading to imminent death, together called “hypovolemic shock complex”. Features on CT include dense opacification of the right heart and major systemic veins, venous layering of contrast material and blood, densely opacified parenchyma in the right hepatic lobe, decreased enhancement of the abdominal organ, a dense pulmonary artery, contrast pooling in dependent lungs, and contrast stasis in pulmonary veins. These findings are biomarkers and prognostic indicators of paramount importance which stratify risk and improve patient outcomes. In this review, we illustrate the various CT patterns in shock and review the spectrum and prognostic significance of thoraco-abdominal vascular and visceral alarming signs of impending death with the intention of increasing awareness among radiologists and radiographers to prepare for immediate resuscitation when required.

## 1. Introduction

Shock is a life-threatening condition defined as a state of cellular and tissue hypoxia due to reduced delivery, increased consumption, inadequate utilization of oxygen, or a combination of these processes [1,2]. 

Shock is not a disease but a continuum of systemic derangement and the final manifestation of a complex list of etiologies [3].

The four different subgroups of shock with characteristic hemodynamic patterns assigned to four organ systems include: distributive shock (vascular system), hypovolemic shock (blood and fluids compartment), cardiogenic shock (heart), and obstructive shock-(circulatory system) (Table 1) [4,5,6,7,8,9,10,11]. It is important to distinguish between these entities since treatment is different for various underlying etiologies [12].

Initially, the effects of shock are reversible as the body initiates compensatory responses to counteract diminished tissue perfusion. However, if the underlying cause is not effectively addressed, shock can progress to irreversible multi-organ failure and ulti-mately result in death [1,3,4,6,11,12].

When patients with undifferentiated hypotension or shock arrive at the emergency department (ED), it is crucial for the emergency physician to categorize them based on the severity of shock and determine the requirement for immediate or early intervention.

Contrast-enhanced multidetector CT (CECT) of the chest, abdomen and pelvis is increasingly required as the first line of imaging in suspected cardiovascular emergencies in severely sick and unstable patients [13]. CT may help identify the cause for shock (e.g., obstructive shock in pulmonary embolism or aortic dissection) [14]. Previous reports have predominantly focused on describing distinct imaging signs observed on CT scans, commonly referred to as the “hypovolemic shock complex” (HSC) [15,16,17,18,19,20,21,22,23,24,25,26,27,28,29,30,31,32]. While these findings are frequently observed in patients with hemorrhagic and hypovolemic shock complex, they can also be present in individuals experiencing conditions such as myocardial infarction, sepsis, or even diabetic ketoacidosis [33]. Recognizing HSC findings provides valuable biomarkers and prognostic indicators, enabling effective risk stratification and improved outcomes for patients in shock. In this review article, we discuss the utilization of CT in vascular and visceral/solid organ shock and describe the various findings, providing an organ-by-organ review of the HSC pattern. Each depicted sign represents a probability and all findings can collectively be used to diagnose, stratify risk of mortality, and guide future clinical management. To optimally manage patient care, radiologists should be familiar with accurate interpretation of these urgent CT findings to inform the responsible clinical physicians for prompt management of these patients.

## 2. Multidetector CT (MDCT) Technique

Detection of relevant CT signs must be accurate, reproducible, and feasible over time, for which a state-of-the-art CECT technique is needed. Nowadays, CT technology consists of a multidetector-spiral CT between 8- and 640-slice CT. Acquisition times and number of contrast-enhanced phases are not standardized for all CT scanners.

CECT should be performed in critical patients with a volumetric technique, with craniocaudal acquisition, in a supine position, and, when possible, preferably in a “feet first” position [34]. Breath-hold acquisition can ensure the avoidance of motion artifacts if clinical condition permits [35]. The CECT multiphasic protocol should include an abdomen/pelvis pre-contrast scan followed by chest/abdomen/pelvis dynamic images acquired in the arterial and portal venous phases without oral contrast medium (CM). CM volume should be calculated according to patient size based on total body weight (BW), injecting about 0.625 gI per kilogram of total BW. High concentrations (370–400 mg I/mL) of IV CM (80–120 mL of iodinated CM, depending on the patient’s weight) should be administered through an 18–20-gauge needle into the antecubital vein at a rate of 3.5–4 mL/s. This should be followed by a bolus of 30–40 mL of saline at the same flow rate. The acquisition of the arterial phase is timed using bolus tracking, placing the region of interest (ROI) on the aortic arch and starting at an attenuation threshold of 100 Hounsfield Units (HU) [34,35,36,37]. The portal venous phase is acquired with a delay of 60–70 s. The suggested acquisition includes scanning the abdomen and pelvis in the arterial phase, and the chest, abdomen, and pelvis in the portal venous phase. Additionally, a late scan of the abdomen and pelvis at 3–5 min may be acquired to address various causes of shock. Oral or rectal CM is not recommended. Each institution should regularly assess image quality, review protocols regarding dose, and consider the possibility of reducing the quantity of CM [34]. The development of new technologies aims to reduce radiation exposure while maintaining good image quality through iterative reconstruction or automatic tube current modulation [35]. Another option to reduce the radiation dose is the adoption of dual-energy CT, allowing the possibility of virtual noncontrast (VNC) image acquisition [38]. An unenhanced CT brain should be considered for patients presenting with altered mental status, to exclude the presence of acute ischemic stroke or intracranial hemorrhage. Head CT can also be conducted during the late phase of a total body study to rule out the presence of intracranial abscess formation or malignancy [35]. To ensure proper analysis and post-processing, it is recommended to use an effective slice thickness of 2.5 mm with reconstruction at 0.625 mm, allowing for maximum intensity projection (MIP) and multiplanar reformation (MPR) techniques.

## 3. CT Patterns

CECT images may be used to assess three possible hemodynamic instabilities in acutely sick patients:(a)In cases of hemodynamic stability, IV CM into an upper limb vein is delivered to the right atrium via the superior vena cava (SVC), and is then pumped via the right ventricle to the pulmonary arteries. Contrast subsequently returns via the pulmonary veins to the left-side cardiac chambers before reaching systemic circulation [39]. As it undergoes first pass circulation and re-circulation, the contrast bolus gradually mixes with the blood pool, leading to dilution while moving downstream from the injection site. Due to its small molecular size, iodinated CM exhibits high diffusibility, readily redistributing from the intravascular space to organic interstitial spaces [39,40]. This may be called the “physiological” pattern and can correspond to an early compensatory stage of shock. Particularly in these patients without advanced shock symptoms, an image-based morphological indicator promises information about the identification of patients “at high risk”.(b)In a state of advanced hemodynamic instability, many homeostatic mechanisms try to maintain arterial pressure and adequate tissue perfusion to critical organs, such as the brain and heart, by reflex stimulation of the sympathetic nervous system, elevated levels of angiotensin II, adrenaline, and noradrenaline, and vasoconstriction (compensated shock). Carotid baroreceptors respond to decreased blood pressure by triggering increased sympathetic signaling and maintaining cardiac output (sympathetic “fight or flight” response). In cases of decompensated shock, when compensatory mechanisms falter and prior to the onset of death, the pumping action of the heart ceases, leading to a substantial decline in systemic arterial and venous pressures. Consequently, the arteriovenous pressure gradient diminishes [6,41,42]. This altered hemodynamic state results in stasis of CM in the venous system in the presence of the left chamber and arterial opacification, and of other infrequent and often unappreciated ominous MDCT vascular signs that represent a true hypovolemic state and must be recognized early by the radiological staff to improve survival [24,43,44,45,46,47,48]. This may be called the “venous CM pooling and layering” pattern, indicating that compensatory mechanisms are becoming insufficient and the patient must receive immediate treatment.(c)In irreversible end-organ dysfunction, injected IV CM circulation is supported only by the pressure applied by the automated power injector and the density of contrast material. Circulatory arrest leads to dense contrast pooling and layering in the SVC, IVC (inferior vena cava), and right heart chambers with non-opacified left heart chambers or arterial vessels (Figure 1) [43,45,49,50,51,52]. This may be called the “non-beating heart” pattern. Cardio-pulmonary aggressive resuscitation must immediately be initiated within the framework of a predetermined emergency plan.

## 4. CT-Updated HSC Findings as Diagnostic Biomarkers

CECT shock-associated findings partially overlap with those referred to previously in the literature as CT “hypoperfusion or hypovolemic shock complex” (HSC) [15,16,17,18,19,20,21,22,23,24,25,26,27,28,29,30,31,32]. This refers to a constellation of findings that reflect hypovolemia and is often described in traumatic hemorrhagic shock [15,16,18,21,24,25,26,27,29]. HSC findings can be grouped into vascular (morphological and functional) and, based on their various anatomic locations, visceral/solid organ findings. It is now clear that vascular signs represent the true hypovolemic state and visceral findings represent hypoperfusion [30].

### 4.1. Vascular Findings (Representing Hypovolemic State)

#### 4.1.1. Morphological Reduction of Vessel Caliber

There is a densely enhanced small-caliber abdominal aorta (with a reduced antero-posterior diameter < 13 mm, detected 20 mm above and below the renal arteries). This finding is commonly associated with hypovolemia, resulting from the arterial vasoconstrictive effects of angiotensin II. It is a non-specific finding observed in approximately 30% of cases of hypovolemic shock, whether traumatic or non-traumatic. It is important to note that this finding can also be observed in the normal population [15,16,17,18,19,20,21,22,23,24,25,26,27,28,29,30,31,32,53,54,55,56].There is the presence of a slit-like or flat inferior vena cava (FIVC) (Figure 2A). This is characterized by an anterior-posterior diameter of less than 9 mm in three consecutive segments, 20 mm above and below the renal veins, and at the level of the perihepatic region. Additionally, a transverse-to-anteroposterior ratio of ≥2.5 at the level of the suprarenal IVC can indicate flattening. The flatness index or IVC diameter ratio is calculated by dividing the maximal transverse and anteroposterior diameters of the IVC [15,16,17,18,19,20,21,22,23,24,25,26,27,28,29,30,31,32,57,58,59,60,61].Flattening of the IVC (slit sign) is often seen in cases of decreased circulating blood volume (hypovolemia) and indicates reduced venous return in patients with systemic hypotension. However, it may not be easily appreciated due to the administration of large volumes of fluids [31]. This finding is more commonly observed in acute hypovolemic traumatic patients. Variations in intra-abdominal pressure and the respiratory cycle can also affect the diameter of the IVC. IVC flattening has a specificity of 90% and a sensitivity of 84% in identifying hypo-perfusion shock in spontaneously breathing patients [15,16,17,18,19,20,21,22,23,24,25,26,27,28,29,30,31,32,57,58,59,60,61,62,63]. The IVC diameter ratio measured via CT scans can help predict in-hospital mortality in septic shock patients, with a cut-off value of ≥1.3 cm having 75% sensitivity and 42% specificity [63]. It is also useful in determining the amount of blood transfusion required and assessing the volume status of patients with blunt torso trauma. [64].The IVC halo sign is characterized by a low attenuation band (<20 HU) encircling the collapsed intra- and retrohepatic inferior vena cava. This band is caused by a ring or rim of edema [65,66,67]. In cases of severe hypovolemia, approximately 80% of patients may exhibit this sign, resulting from the loss of precapillary arteriolar sphincter tone and the accumulation of fluid surrounding the IVC (Figure 2B–D) [18,25,30,31,65,66,67]. However, it is important to note that this sign is not specific to non-traumatic patients and can also be observed in conditions such as liver congestion, biliary cirrhosis, hepatitis, or other diseases that obstruct lymphatic drainage at the porta hepatis [25].Narrowing of superior mesenteric vessels. In cases of hypovolemic shock, narrowing of the superior mesenteric vessels (diameter less than half that of the aorta and IVC), accompanied by intense enhancement similar to the aorta and IVC, is frequently observed, with a frequency ranging from 88.5% to 96.2% [18,19,20,68]. Splanchnic hypoperfusion can be observed in both hypovolemic shock and non-occlusive mesenteric ischemia (NOMI) and is often attributed to reduced cardiac output and cardiogenic shock.

#### 4.1.2. Functional

Dependent CM pooling and layering/reflux of CM/stasis of CM

In a normal physiological state, specific gravity has no effect on contrast agent dynamics. In heart failure, CM does not mix with the blood pool and its distribution in the vascular system predominantly depends on its density, injector pressure, specific weight and volume injected [21,39,44]. The dependent layering of injected iodinated CM mainly reflects its higher specific gravity relative to blood in cases of right heart dysfunction and very low cardiac output [47,52]. The consequence is reduced enhancement of the aorta and left-side cardiac chambers with reflux of contrast into the IVC, creating a characteristic horizontal blood-contrast level (“dependent pooling sign” or “IVC contrast level sign”) that could be associated with reflux into the hepatic veins with heterogeneous liver parenchyma enhancement (Figure 3) [18,44,52,66,67].

In contrast to horizontal levelling in the IVC layering sign, a vertical IVC levellng sign usually occurs due to physiological retrograde filling of renal veins [68]. Contrast stasis in the right-side cardiac chambers may result in extremely dense chambers and pulmonary arteries with or without a blood contrast level. Vascular stasis will also result in contrast layering within the veins that eventually drain into the right atrium (e.g., brachiocephalic and subclavian veins) (Figure 4).

Focal hot spot sign

Contrast stasis in the right-side cardiac chambers and SVC may result in functional flow of CM through venous collaterals (anterior intercostal, internal thoracic, superior and inferior epigastric veins communicating with paraumbilical vein carrying the blood and the CM to the hepatic vein and to the left lobe of the liver) to the inferior vena cava, generating areas of focally increased blood flow to the liver, typically within segment IV of the left hepatic lobe, known as a focal hot spot sign (Figure 5) [69,70,71].

Hypoattenuating periportal halo

This finding is mainly described in hypovolemic, often blunt traumatic shock as a consequence of large volumes of resuscitating IV fluids. This is depicted as a circumferential region of low attenuation around the intrahepatic portal vessels [72,73,74,75].

Ongoing hemorrhage

In cases of hemorrhagic shock, whether traumatic or non-traumatic, active arterial bleeding is characterized by the presence of linear irregularities or areas of increased intensity “blushes” within or adjacent to the injured organ or artery. These manifestations tend to increase in size during later phases of imaging. Differentiating active bleeding from pseudoaneurysm formation can be achieved by observing the lobular margins and enhancement of the pseudoaneurysm after blood pooling during the arterial and subsequent phases of imaging [76].

### 4.2. Visceral/Solid Organs Findings (Representing Hypoperfusion State)

#### 4.2.1. Thyroid

Thyroid findings are an uncommon, largely unknown part of the CT hypotension complex [19,21,25,26]. To our knowledge, only eight cases have been reported in the literature [77]. In their initial description, Brochert et al. observed a distinct pattern of heterogeneous hyperenhancement in the thyroid gland, resembling the appearance of a multinodular gland. Additionally, surrounding fluid accumulation, referred to as “shock thyroid” was noted, despite the absence of any apparent direct or indirect thyroid injury [78]. The presumed mechanism is that hypoperfusion of the highly vascular thyroid gland may cause cellular edema or death as well as exudation of intracellular fluid. Other proposed explanations include third-spacing of resuscitative fluid and a profound thyroid response, which induces transient thyrotoxicosis and a swollen gland for the maintenance of cardiac output [25,77,78,79]. These glandular changes are reversible when associated with successful management of hypovolemia and hypotension [77].

#### 4.2.2. Lungs

Alteration of the blood circulation dynamics may cause IV contrast to gravitate to the dependent lung segments, resulting in extremely high attenuation of the lungs posteriorly. Bilateral diffuse and lobular Ground Glass Opacification/consolidation, mainly in the middle and lower lung zones with or without an air bronchogram, may be seen [80]. Bilateral pleural effusions and lower lobar passive atelectasis are often ancillary signs of acute right-side heart failure. Acute pulmonary dysfunction and new bilateral infiltrates on chest imaging are common in septic shock from diffuse alveolar epithelial injury, leading to capillary leaks and acute respiratory distress syndrome (Figure 6A,B).

#### 4.2.3. Bowel (Marked Submucosal Edema and Intense Mucosal Enhancement)

Hypovolemia triggers the sympathetic system, leading to splanchnic vasoconstriction and reduced blood perfusion to the bowel. However, mucosal perfusion is preserved through autoregulation mechanisms that prevent ulceration. This results in prominent mucosal enhancement, which appears greater than the psoas muscle on non-contrast images, and submucosal enhancing edematous wall thickening (bowel wall > 3 mm) [21,27]. Severe hypotension can lead to inadequate oxygen delivery to the organs [22,28]. Reduced perfusion can cause injury to the intramural vessels, resulting in increased capillary permeability and interstitial fluid leakage into the bowel wall and lumen. This leads to decreased fluid reabsorption and eventually ileus, characterized by dilated fluid-filled loops (Figure 6C,D) [19,21,81,82,83,84]. Diffuse bowel ischemia due to vascular occlusion or non-occlusive mesenteric ischemia (NOMI) presents a challenging differential diagnosis. Both conditions can cause bowel wall thickening and luminal distention. However, arterial occlusion-related bowel ischemia does not exhibit diffuse mucosal enhancement or substantial submucosal edema. On the other hand, mesenteric venous occlusion or reperfusion in NOMI may exhibit both of these CT signs [82,83,84].

#### 4.2.4. Spleen

The size of the spleen can vary among individuals, but it is typically around 10 × 6 × 3 cm³ in dimensions, weighing approximately 120 g. The spleen is highly vascularized and can store 20–30% of the total blood volume [85]. Unlike other organs, the splenic arterial flow lacks autoregulatory mechanisms and is highly sensitive to overstimulation of the sympathetic system and vasoconstriction. This can lead to a decrease in splenic blood flow and hypoenhancement, typically at least 20 HU less than the liver (Figure 7A,B) [85,86,87].

In cases of hypovolemia, another sign to observe is a reduction in splenic volume. This reduction is attributed to the contraction of smooth muscle in the vessel walls and splenic capsule, which forces red blood cells and platelets to be released from the spleen and into the general circulation. This process is often referred to as “auto-transfusion”. Furthermore, in the setting of shock, particularly heart failure, both splenic volume and splenic volume index are significantly lower (with values of 118.0 mL and 68.9 mL/m^2^, respectively) [88]. Hypoxia and exercise can independently trigger splenic contraction, resulting in the release of stored erythrocytes [89,90].

#### 4.2.5. Liver and Gallbladder

Delayed and heterogeneous liver enhancement, attributed to the liver’s dual blood supply, is a well-documented and commonly referenced sign of shock associated with heightened sympathetic activity. This phenomenon is described in the literature and has been recognized as a characteristic manifestation of shock [18,19,20,21,22,23,91,92]. Increased parenchymal capillary permeability may result in the accumulation of contrast into the dependent right liver parenchyma, causing a gravitationally dense parenchymal appearance [18,19,20,21,22]. A demarcation line between the dependent enhancing and non-dependent non-enhancing parenchyma may be seen and corresponds to the height of the feeding vein (Figure 7C). Abnormal gallbladder mucosal enhancement without wall thickening may also be observed in hypovolemic shock [19,31,93] (Figure 7D).

#### 4.2.6. Adrenals

The presence of intense and persistent symmetric adrenal enhancement, commonly referred to as “shock adrenals”, was initially described in 1992 within the context of hypovolemic shock in the pediatric population. It is characterized by bilateral and symmetric avid enhancement of the adrenal glands during the portal venous phase, exceeding that of the adjacent inferior vena cava [94]. This phenomenon can also occur in adults due to blood flow redirection to the adrenal glands in cases of hypotension. It is a result of reflex stimulation of the hypothalamus–pituitary axis and sub-sequent sympathetic overactivity, leading to elevated levels of noradrenaline and angiotensin II. The presence of intense adrenal enhancement is considered a poor prognostic factor associated with a high mortality rate (as depicted in Figure 7A–C). It can be one of the earliest CT signs of cardiogenic shock [26,95,96,97,98,99].

The hollow adrenal gland sign is specific and common on dual-phase contrast-enhanced CT in 30% of patients with septic shock and predicts poor prognosis [100]. Winzer et al. reported that the portal venous adrenal-to-spleen ratio (opposite enhancement of the adrenal glands (↑) and spleen (↓) on portal venous CT scans, with a cutoff value of 1.37) serves as a reproducible image-based prognostic marker with high predictive power (sensitivity: 83.7%; specificity: 99.1%; positive predictive value: 93.2%; negative predictive value: 97.6%) for short-term (72 h) mortality in ICU patients [101,102].

#### 4.2.7. Pancreas

In cardiogenic shock, glandular enhancement can be increased (20 HU greater than the liver and spleen) or decreased in relation to sympathetic overactivity (Figure 8A,B) [25,103,104]. If this self-regulating system of blood flow fails, reduced perfusion of the pancreas ensues, indicating a state of irreversible shock [19]. Hyperenhancement of the adrenals may also be observed in traumatic hypovolemic shock, such as large surface and chemical burns. Peripancreatic fluid in the absence of pancreatitis, pancreatic disease or traumatic injury to the pancreas is secondary to increased pancreatic permeability [19,21].

#### 4.2.8. Stomach

Sympathetic activation exerts a predominantly inhibitory effect on gastric muscle and provides a tonic inhibitory influence on mucosal secretion, while at the same time regulating gastrointestinal blood flow via neurally mediated vasoconstriction [105]. We present cases of septic/cardiogenic shock with a massively distended, fluid-filled hypotonic stomach, with poor thin wall enhancement in early and late arterial phases, but preserved parietography in the delayed phase (shock stomach) (Figure 8C,D). To the best of our knowledge, this finding was not described previously in the literature.

#### 4.2.9. Kidneys

Renal perfusion abnormalities commonly present as heightened and prolonged parenchymal enhancement, often referred to as “white kidneys”. Nonetheless, it is important to note that focal and heterogeneous enhancement patterns may also be detected. After an early period of intense prolonged enhancement, the kidneys may progress to non-enhancement if the patient’s condition continues to deteriorate. Absent enhancement of the kidneys may cause “black kidney sign” [106]. After an acute fall in systolic pressure and sympathetic stimulation, efferent glomerular arteriole vasoconstriction develops with renal parenchymal CM stasis, resulting in prolonged cortical hyperenhancement of the kidneys [19,20]. It is also associated with absent enhancement or hypoenhancement of the renal medulla on delayed phase image acquisition, attributable to acute tubular necrosis, which is a poor prognostic sign in the presence of a hypovolemic state [19,20,22]. Persistent bilateral nephrograms are characteristically and usually distinguished from delayed nephrograms, which occur unilaterally. However, renal enhancement must be evaluated carefully as a sign of hypovolemia, as the timing of contrast injection can affect the appearance of renal enhancement [26].

#### 4.2.10. Ascites

Ascites is a non-specific sign that correlates with the underlying etiology and multiorgan dysfunction.

## 5. CECT Findings/Biomarkers as Prognostic Indicators

The clinical diagnosis of hypovolemic shock is not always obvious in an acute setting due to hemodynamic compensatory mechanisms. Most of the published literature refers to hypovolemic shock, and cardiogenic shock to a lesser extent, often in a trauma setting. There is a scarcity of radiological literature regarding CT findings in distributive and obstructive shock. CT findings are biomarkers that can be used to predict the diagnosis and prognosis of shock as well as being useful for monitoring the response to treatment. While specific CT signs in different subtypes of shock depend on the underlying etiology(ies), CT features generally overlap all forms. To confirm the diagnosis of CT shock syndrome, the presence of two or more vascular, visceral, or parenchymal signs is required [18,27]. The incidence of CT findings as prognostic indicators in various shock types is summarized in Table 2.

## 6. Conclusions and Future Directions

Patients in shock require focused management without delay in order to improve morbidity and mortality. In the right clinical context, various imaging signs can be utilized to stratify urgency and direct future management. Radiologists need to acquaint themselves with imaging findings associated with different types of shock and become familiar with several factors limiting interpretation. In addition, radiologists should readily communicate their findings to the responsible physicians in order to provide the best quality care in a timely manner.

Future development of technologies (e.g., photon counting CT) aiming to reduce artifacts and improve resolution will undoubtedly promote the superior delineation of acute findings related to different states of shock while minimizing patient radiation exposure.

## Figures and Tables

**Figure 1 diagnostics-13-02304-f001:**
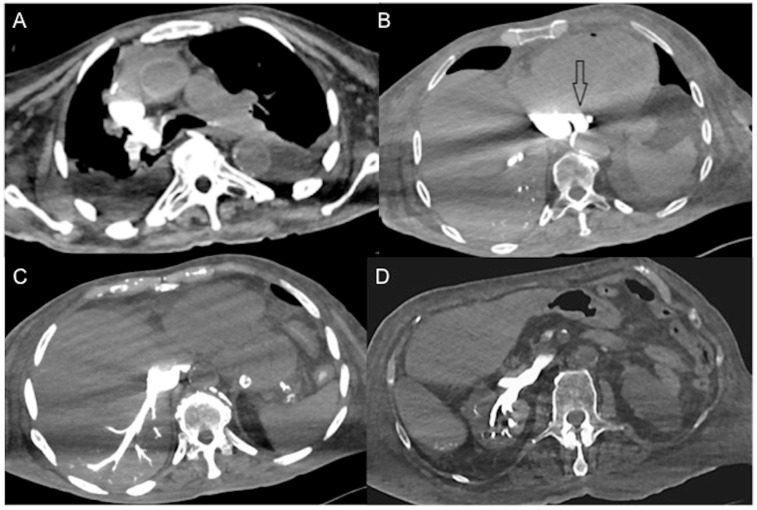
Non-beating heart in a 72-year-old man with sudden-onset severe dyspnea/shock and asystole during thoraco-abdominal CT. (**A**) CECT axial image shows dense contrast in the round superior vena cava, and reflux in the azygous arch; (**B**) contrast pooling and layering in the right atrium and IVC with retrograde opacification of coronary sinus (arrow). (**C**) CM fills the round inferior vena cava with hypostatic reflux into the hepatic veins, hemiazygos vein, partially splenic vein, and (**D**) right renal vein. Note no mixing of blood with CM and no opacification of the pulmonary arteries, aorta, and left cardiac chambers, suggestive of a non-beating heart. Prompt initiation of cardio-pulmonary resuscitation to restore circulation was useless. Autopsy: ruptured myocardial infarction.

**Figure 2 diagnostics-13-02304-f002:**
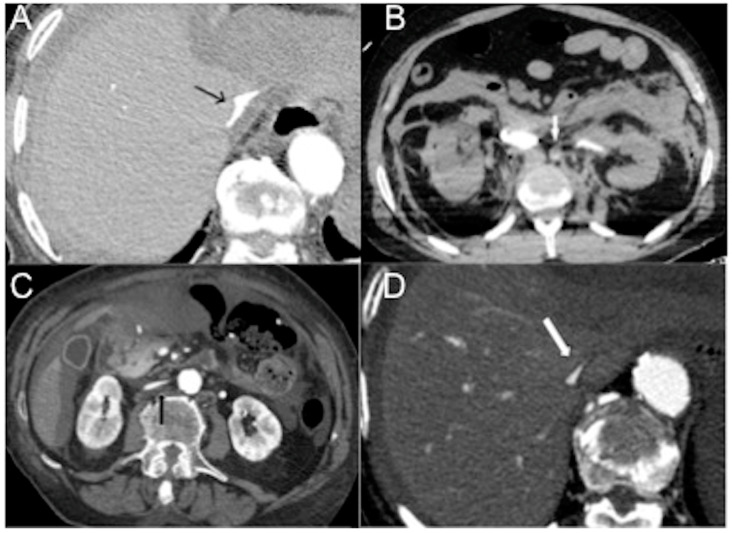
CECT axial images of vascular findings in the four subtypes of hypovolemic shock. (**A**) Slit-like IVC (arrow) in massively bleeding enteric infarct (hemorrhagic non-traumatic shock). (**B**) Small caliber hypoenhanced aorta (arrow) in a 32-year-old man wounded by several gunshots, who died twenty minutes after CT examination (traumatic hemorrhagic shock). The characteristic feature of hemorrhagic and traumatic hemorrhagic shock is bleeding. (**C**) Flat IVC (black arrow) and ascites in a 57-year-old man with decompensed liver cirrhosis and critical reduction in circulating plasma volume (hypovolemic shock in the narrower sense). (**D**) Flat IVC with halo sign (white arrow) in a 45 year-old firefighter burned with lung toxicity and significant fluid loss (traumatic hypovolemic shock).

**Figure 3 diagnostics-13-02304-f003:**
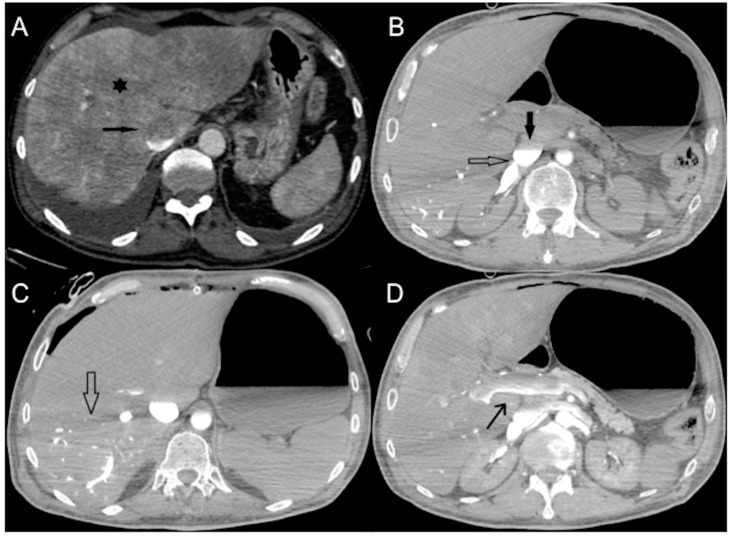
CECT axial images of vascular findings in cardiogenic shock. (**A**) Portal venous phase (delay of 80 s after CM injection) shows IVC contrast level sign (arrow) as well as heterogeneous hepatic enhancement (asterisk) in a 65-year-old man with myocardial infarction and imminent cardiac arrest. (**B**) Portal venous phase (delay of 85 s after CM injection) shows IVC contrast level sign (arrow) and regurgitation to the right renal vein (empty arrow) in a 63-year-old man with hypotension and acute severe reduction in cardiac index. In comparison to Figure 1D, note the aorta opacification. (**C**) Portal venous phase (delay of 85 s after CM injection) shows IVC contrast level sign with CM reflux into hepatic VI and VII segments (empty arrow) and (**D**) contrast level in portal trunk (arrow) in a 68-year-old man with myocardial infarction.

**Figure 4 diagnostics-13-02304-f004:**
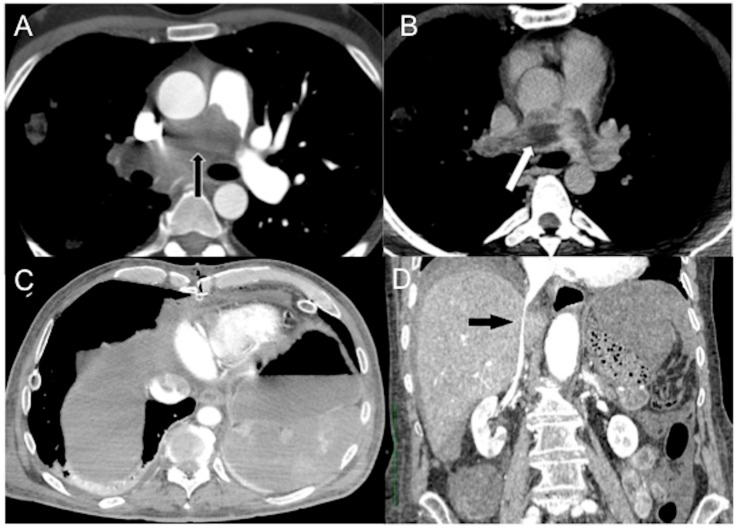
CECT images in obstructive shock in a severely dyspneic 28-year-old man with malignant pulmonary artery sarcoma. Arterial (**A**) and delayed (**B**) phase axial images show massive pulmonary trunk (black arrow) and right main artery (white arrow) soft tissue mass obstruction. A more caudal (**C**) axial image shows CM stasis in right cardiac chambers, no hepatic enhancement, and mild pleuropericardial effusion. (**D**) Delayed phase (delay of 200 seconds after CM injection) coronal reconstruction shows a thinned flat IVC (black arrow).

**Figure 5 diagnostics-13-02304-f005:**
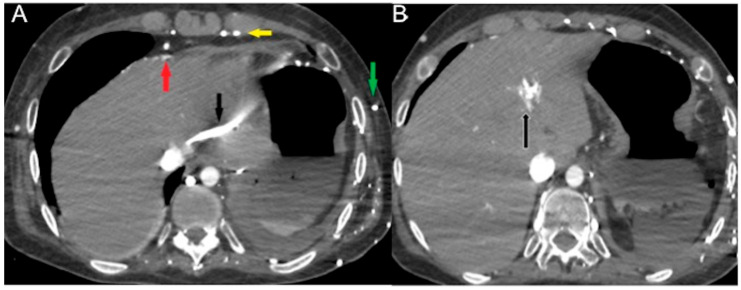
CECT axial images in an acutely dyspneic 64-year-old man with SVC obstruction from right lung cancer (not shown) and obstructive shock. Arterial phase (**A**) shows CM reflux via the larger subcutaneous (green arrow), pericardiophrenic (red arrow) and internal thoracic veins (yellow arrow) to the left hepatic vein (black arrow) generating (**B**) focal hot spot sign in left liver (black arrow).

**Figure 6 diagnostics-13-02304-f006:**
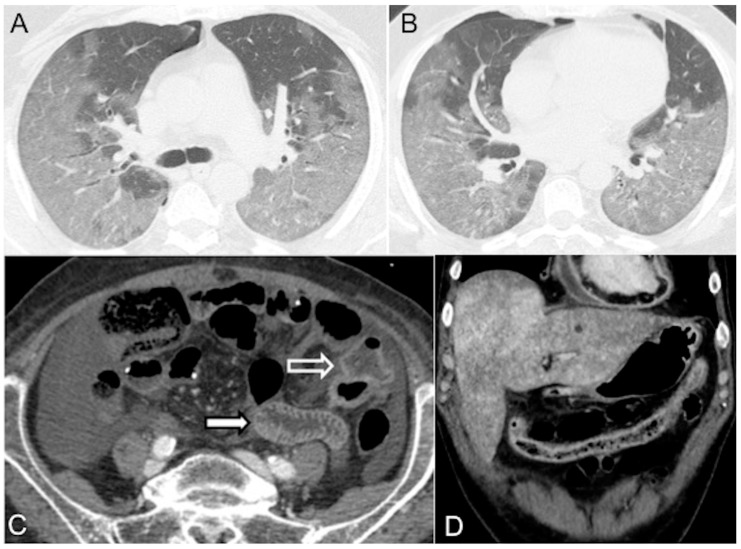
Distributive septic shock in a 61-year-old male admitted to the emergency department with high fever (39 °C) and sepsis (score for sepsis, qSOFA 3) due to COVID-19 pneumonia (respiratory rate 27/min, systolic blood pressure 75 mmHg, altered mental status). (**A**,**B**) Unenhanced chest CT axial images (lung window) show bilateral parenchymal ground-glass opacities due to a diffuse alveolar damage (DAD) pattern; note mild right pneumothorax. (**C**) CECT axial image in the portal venous phase shows mural thickening (white empty arrow) and mucosal hyperenhancement of the small bowel (white arrow). (**D**) CECT coronal reconstruction in the portal venous phase shows abnormal wall thickening/enhancement in the partially collapsed transverse colon. Note associated heterogeneous liver enhancement and enhanced thickened pericardium with pericardial effusion (pericarditis).

**Figure 7 diagnostics-13-02304-f007:**
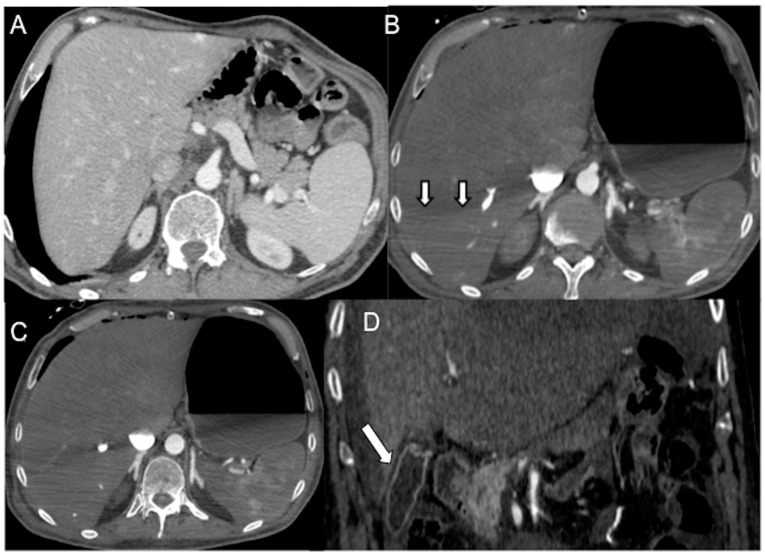
Pre-surgical (**A**) and first day after thoracic surgery (**B**,**C**) CECT axial images in a 65-year-old man with acute septic shock. (**A**) The portal image shows normal splenic volume and enhancement, physiologic size and enhancement of adrenals and a little sub-capsular angioma in the left liver. (**B**) Arterial phase image of emergent CECT shows splenic volume contraction and hypoperfusion. A demarcation line between the dependent/enhancing and non-dependent/non-enhancing liver parenchyma is present (white arrows). Note hyper-enhancing adrenals and thin-walled luminal distended stomach. (**C**) The 85″ venous phase shows a persistent hypoenhanced liver, adrenals hyperenhancement and stomach parietogram. (**D**) In a 57-year-old man with decompensed liver cirrhosis and critical reduction in circulating plasma, CECT coronal reconstruction shows dense gallbladder mural enhancement without thickened walls (white arrow).

**Figure 8 diagnostics-13-02304-f008:**
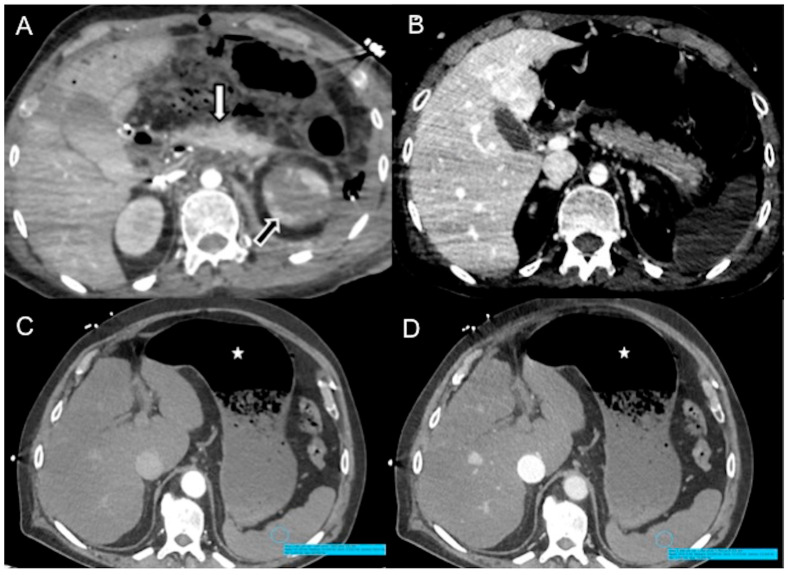
CECT and variable pancreatic enhancement. (**A**) Axial image shows increased pancreatic enhancement (white arrow) and hypoenhancing left kidney (cortical enhancement preserved) (black arrow) in traumatic hemorrhagic shock (multiple thoraco-abdominal gunshot wounds); note flat IVC. (**B**) Axial image shows decreased pancreatic enhancement in distributive neurogenic shock in a 69-year-old-man with acute basilary artery occlusion. Note non-enhancing spleen and liver heterogeneous enhancement (hot spot sign in left liver). Axial images show splenic hypoenhancement in arterial (**C**) and venous portal (**D**) phases (blue circle) in distributive septic shock. Note distended fluid-filled hypotonic stomach (star) with poor thin wall enhancement in arterial phase, but preserved parietography in the delayed phase.

**Table 1 diagnostics-13-02304-t001:** Classification (causes, pathogenesis, and treatment targets) and relative incidence of various types of shock. Systemic arterial hypotension, cutaneous, renal, and neurological signs of tissue hypoperfusion and hyperlactatemia are often present in all shock pathophysiological mechanisms.

Shock				
Types	Pathogenesis	Causes	Pathophysiology	Treatment Targets
Cardiogenic (13%)	Sudden impairment of myocardial performance	Myocardial Rhytmologic Mechanical	A critical reduction of the heart’s pumping capacity, a reduced ejection fraction or impaired ventricular filling	Remove the cardiac causes of the shock
Hypovolemic (27%)	Inadequate organ perfusion caused by loss of intravascular volume	Hemorragic (e.g., variceal bleeding) Traumatic hemorragic Hypovolemic in the narrower sense (e.g., diarrhea) Traumatic hypovolemic (e.g., severe surface burns)	Inadequate organ perfusion caused by acute loss of intravascular volume, drop in cardiac preload to a critical level	Intravascular volume replacement, endotracheal intubation
Distributive (59%)	Hypovolemia resulting from pathological redistribution of the absolute intravascular volume	Septic (55%) Anaphylactid/Anaphylactoid Neurogenic Systemic inflammatory response syndrome (SIRS) Drug and Toxin-induced shock Endocrine shock	Loss of regulation of vascular tone and/or disordered permeability of the vascular system	Support circulation by infusion of balanced solutions, administration of vasopressors and/or inotropic drugs, organ replacement therapy
Obstructive (1%)	RV-LV Preload ↓ RV-LV Afterload ↑ Obstruction of the great vessels or the heart	Classified according to the location of the obstruction in the vascular system in relation to the heart (e.g., SVCs., PE, Obstruction aortic flow)	Intravasal/Intraluminal (e.g., PE, Leriche S., AD) Extravasal/extraluminal (e.g., Tension PNX, Tamponade)	Immediate causal treatment (e.g., thrombolysis, thoracic or pericardial drainage; surgical embolectomy)

AD, aortic dissection; LV, left ventricle; PE, pulmonary embolism; PNX, pneumothorax; RV, right ventricle; SVC, superior vena cava. ↓: reduce, ↑: increase.

**Table 2 diagnostics-13-02304-t002:** Incidence and prognostic meaning of CECT findings in each shock type (from references [15,16,17,18,19,20,21,22,23,24,25,26,27,28,29,30,31,32,43,44,45,46,47,48,49,50,51,52,53,54,55,56,57,58,59,60,61,62,63,64,65,66,67,68,69,70,71,72,73,74,75,76,77,78,79,80,81,82,83,84,85,86,87,88,89,90,91,92,93,94,95,96,97,98,99,100,101,102,103,104,105,106]). The presence of two or more vascular, visceral, or parenchymal signs is deemed necessary to establish the presence of CT shock syndrome. IVC, Inferior vena cava; NA, not available; SMA/V, Superior mesenteric artery/vein; AV, Atrio-ventricular.

CECT Findings	Cardiogenic	Distributive	Hypovolemic	Obstructive	Outcome
Small-caliber aorta	~25%	~28%	~30%	~35%	poor
Slit/flattened cava	~70%	~55%	~77%	~50%	very poor
Halo sign IVC	~70%	~65%	~75%	NA	poor
Narrow SMA/V	NA	NA	NA	NA	NA
Lack of left AV enhancement	~65%	~35%	~55%	~20–50%	very poor
CM vascular layering	~75%	NA	~65%	~70%	very poor
Hot-spot sign	NA	NA	NA	NA	NA
Periportal halo	~60%	NA	~40%	NA	NA
Ongoing hemorrhage	10%	15%	65%	25%	poor
Shock Thyroid	NA	NA	NA	NA	very poor
Shock Lungs	NA	NA	NA	NA	NA
Shock Bowel	~55%	~50%	~70%	~40%	poor
Shock Spleen	~40%	~50%	~50%	~25%	poor
Liver altered density	~85%	~55%	~57%	~45%	poor
Shock gallbladder	20–30%	~12%	13–35%	~9%	poor
Shock pancreas	~35%	~55%	~45%	~35%	very poor
Shock Stomach	NA	NA	NA	NA	NA
Shock Kidneys	~55%	~50%	~60%	~40%	poor
Shock Adrenals	~60%	~65%	~55%	~50%	poor

## Data Availability

Not applicable.

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
