# Peer review of "Multidetector CT Imaging Biomarkers as Predictors of Prognosis in Shock: Updates and Future Directions"

_diagnostics, 2023, doi:10.3390/diagnostics13132304_

Round 1
Reviewer 1 Report
The paper reviews the role of CT as the predictor of prognosis in shock.
Title: future direction was not adequately described. Remove multidetector, as CT have almost all multidetectors.
Introduction: please do not make a sentence a paragraph.
Table 1; list abbreviations in an alphabetical order.
L66: radiologist
Table 2: please provide percentages only in items supported by references.
Figures: please not use red arrows or circles. Use black arrows with white border or vice versa.
The quality of English is good. However, typographic errors should be checked carefully including the use of capital letters.
Author Response
Please do not see the attachment!
Thanks for reviewer 1 suggestions!
Point 1: Title: a) future direction was not adequately described. b) Remove multidetector, as CT have almost all multidetectors.
Response 1a:
- a) Conclusion and future direction
Patients in shock require focused management without delay in order to improve morbidity and mortality. In the right clinical context, various imaging signs can be utilized to stratify urgency and direct future management. Radiologists need to acquaint themselves with imaging findings associated with different types of shock and become familiar with several factors limiting interpretation. In addition, radiologists should readily communicate their findings to the responsible physicians in order to provide the best quality care in a timely manner.
Future development of technologies (e.g., photon counting CT) aiming to reduce artifact and improve resolution will undoubtedly promote superior delineation of acute findings related to different states of shock whilst minimizing patient radiation exposure.
Response 1b: Multidetector may be removed from the title if in accordance with Reviewer 2's request (that is: Point 2: Line 69; (Section) 2. MDCT technique should be changed to Multidetector CT (MDCT) technique.)
The title can therefore become “CT imaging biomarkers as predictors of prognosis in shock: update and future direction.”
Point 2: Introduction: please do not make a sentence a paragraph.
Response 2: I have shortened the length of all the sentences in the Introduction.
Point 3: Table 1; list abbreviations in an alphabetical order.
Response 3: AD, aortic dissection; LV, left Ventricle; PE, pulmonary embolism; PNX, pneumothorax; RV. right ventricle; SVC, Superior Vena Cava.
Point 4: L66: radiologist (instead of Radiologist)
Response 4: radiologist
Point 5: Table 2: please provide percentages only in items supported by references.
Response 5: I have changed the Table 2 and provided percentages only in items supported by references
Point 6: Figures: please not use red arrows or circles. Use black arrows with white border or vice versa.
Response 6: I have changed all Figures except Figure 5A where the different colors indicate three different collateral venous pathways. For all other figures I used black arrows with white border or vice versa.

Reviewer 2 Report
The manuscript titled " Multidetector CT imaging biomarkers as predictors of prognosis in shock: update and future direction" presents a captivating review article focusing on the hypovolemic shock complex and its CT imaging biomarkers. This systematic review takes readers on a comprehensive journey, starting from the fundamentals and delving into intricate scientific details. It offers an in-depth examination of the hypovolemic shock complex pattern, meticulously covering each organ. Overall, this review is truly remarkable in its scope and content.
1. Table 1; I would write Hypovolemic before Distributive.
2. Line 69; (Section) 2. MDCT technique should be changed as Multidetector CT (MDCT) technique.
3. Line 80; the commonly used term is body weight (BW), not TBW.
4. Line 90; authors should elaborate “Oral or rectal CM is inappropriate”.
5. Line 381; What does author mean “It occurs also in adults because of redirection of blood flow to the adrenal glands in cases of hypotension because of a reflex stimulation of the hypothalamus-pituitary axis and consequent sympathetic overactivity (elevated levels of nor-adrenaline and angiotensin II). This is a poor prognostic factor associated with a high mortality rate.” Authors should clarify.
Quality of English language is good. Minor revision is required to improve grammar and remove typo errors.
Author Response
Please do not see the attachment!
Thanks for Reviewer 2 suggestions!
Point 1: Table 1; I would write Hypovolemic before Distributive.
Response 1: I have changed Table 1 (Hypovolemic shock before Distributive)
Point 2: Line 69; (Section) 2. MDCT technique should be changed to Multidetector CT (MDCT) technique.
Response 2: MDCT technique has been changed to Multidetector CT (MDCT) technique (if in accordance with Reviewer 1's request that is: Remove multidetector from the title, as CT have almost all multidetectors.)
Point 3: Line 80; the commonly used term is body weight (BW), not TBW.
Response 3: Line 80 and 81 total body weight (TBW) has been changed to total body weight (BW)
Point 4. Line 90; authors should elaborate “Oral or rectal CM is inappropriate”.
Response 4: The improved image quality of the multidetector CT scanners and the longer acquisition time with the use of oral or rectal CM of these emergency CT examinations make the administration of these positive CM inappropriate.
Point 5: Line 381; What does author mean “It occurs also in adults because of redirection of blood flow to the adrenal glands in cases of hypotension because of a reflex stimulation of the hypothalamus-pituitary axis and consequent sympathetic overactivity (elevated levels of nor-adrenaline and angiotensin II). This is a poor prognostic factor associated with a high mortality rate.” Authors should clarify.
Response 5: This phenomenon can also occur in adults due to blood flow redirection to the ad-renal glands in cases of hypotension. It is a result of reflex stimulation of the hypothalamus-pituitary axis and sub-sequent sympathetic overactivity, leading to elevated levels of noradrenaline and angiotensin II. The presence of intense adrenal enhancement is considered a poor prognostic factor associated with a high mortality rate (as depicted in Figures 7A-C). It can be one of the earliest CT signs of cardiogenic shock [26,96-100].
